PERSPECTIVE

# Challenges in implementing cultural adaptations of digital health interventions

Vasileios Nittas[1✉], Paola Daniore[2], Sarah J. Chavez[3] & Tyler B. Wray[1]

Differences in the access and use of digital health interventions are driven by culture, in addition to economic and physical factors. To avoid the systematic exclusion of traditionally underserved cultural groups, creating inclusive digital health interventions is essential. One way to achieve this is through cultural adaptations, defined as the systematic modification of an existing intervention that aligns with a target audience's cultural norms, beliefs, and values. In theory, cultural adaptations can potentially increase the reach and engagement of digital health interventions. However, the evidence of whether and how that is achieved is limited. Justifying, planning, and implementing an adaptation comes with various challenges and takes time and money. This perspective provides a critical overview of the field's current state and emphasizes the need for technology-specific frameworks that address when and how to culturally adapt digital health interventions.

The COVID-19 pandemic has shown that unequal access to digital technology including smart phones, tablets, computers, and the internet, collectively known as the digital divide, remains an unsolved public health challenge[1]. During the early phase of the pandemic, many aspects of life, including healthcare delivery, moved to be online-only, which left many excluded due to this digital divide[2,3]. For example, in the USA, 20% of households faced lockdowns without regular access to a broadband internet connection[3]. While many Black and Latino households relied instead on smartphone-only subscriptions, others completely lacked internet connectivity[3]. Although internet access in some public spaces (e.g., schools, outdoor areas, shopping centers) is expanding, privacy concerns might limit its use for healthcare. The use of digital health is also determined by how well each product addresses the needs, circumstances, and worldviews of people, as well as how well it fits into their daily lives[4]. Lack of trust in technology, bias, as well as racism further increase disparities[5,6]. One way to narrow the digital divide is to implement more culturally relevant digital health interventions (DHIs).

Recently, the World Health Organization (WHO)'s European Regional committee adopted an action framework that calls on all Member States to carefully consider behavioral and cultural aspects when developing health policies, including DHIs[7]. One way to create culturally relevant DHIs is through cultural adaptation; that is, the systematic modification of an existing intervention so that it aligns with a target audience's cultural norms, beliefs, and values[8]. The rationale behind cultural adaptions is to improve the reach and engagement of otherwise underserved population subgroups[9]. Reach can be defined as initial engagement, which requires personal interest and trust in an intervention[10,11]. It is ideally followed by continuous engagement, defined as the interaction with an intervention, as prescribed, until it is fully delivered[12]. Yet, little is known about when and how to adapt DHIs to increase reach or engagement for systematically underserved cultural groups.

We base this perspective on the assumption that cultural adaptations of DHIs have the potential to reduce digital divides by increasing the reach and engagement of traditionally

[1]Department of Behavioral and Social Sciences, Brown University School of Public Health, 121 South Main Street, 02912 Providence, RI, USA. [2]Institute for Implementation Science in Health Care, University of Zurich Faculty of Medicine, Universitaetstrasse 84, 8006 Zurich, Switzerland. [3]Center for Alcohol and Addiction Studies, Brown University School of Public Health, 121 South Main Street, 02912 Providence, RI, USA. ✉email: Nittas_vasilios@brown.edu

excluded cultural subgroups. We emphasize the current evidence gap and focus on critical challenges (Box 1) and their potential mitigations (Box 2), with the goal of informing those who aim to culturally adapt existing DHIs (e.g., researchers, software developers). We define DHIs as any structured program using information and communication technology to improve health.

## Culture and the digital divide

In the digital world, neither reach nor engagement is free from inequities[13]. The term digital divide describes the gap between those who have the opportunity and skills (digital literacy) to use technology to their benefit, and those who do not, also referred to as digital inequities[14,15]. In the context of health, digital divides are driven by many factors, including, but not limited to one's built environment, social context, educational background, and economic stability[16]. These factors may influence access to, the use of, and attitudes towards technology[8,15,17,18]. Research undertaken in the USA suggests that cultural minorities, across racial and ethnic subgroups, continue to experience reach and engagement barriers when accessing healthcare services[13]. For example, racial and ethnic minorities are less likely to use certain technologies to manage their health, linked to several socio-cultural factors (e.g. language, education, cultural familiarity)[19]. In such situations, cultural adaptations aim to elicit the interest of traditionally excluded and hard-to-engage cultural groups by providing DHIs that are aligned with an individual group's preferences, beliefs, values, and needs[11]. Yet, culture and the cultural adaptation of DHIs remain largely overlooked parameters of the digital divide[15,17,18].

## The evidence gap

While cultural adaptations of non-digital interventions have often proven to be beneficial, the evidence that culturally adapting DHIs provides benefits is still lacking[8–10,17]. Two recent systematic reviews emphasize this gap[20,21]. Spanhel et al. focused on culturally adapted internet- and mobile-based interventions for mental disorders, summarizing the results of 55 studies[20]. The review identified 17 adaptation criteria, with an average of 12 met

by each included study[20]. Although none of the included studies directly compared the original and adapted interventions. The review reported that in most cases, there was no clear evidence of improved effectiveness, or engagement among DHIs that were culturally adapted[20].

Balci et al. looked at culturally adapted internet- and mobile-based health promotion interventions, summarizing the results of 13 randomized controlled trials (RCTs)[21]. Nine RCTs conducted only surface adaptations, targeting observable and obvious cultural elements such as language[21,22]. The remaining four studies conducted surface and deep content changes[21]. Overall, the study's meta-analysis demonstrated that, except for physical activity interventions, which showed significant long-term effects, the tested culturally adapted interventions were not superior to their controls[21]. The authors concluded that cultural adapted DHIs were not superior to non-adapted ones and thus "might not be worth the effort"[21]. However, we do not believe that the evidence was robust enough to draw such firm conclusions. The number of currently available studies is small, and their chosen outcomes are heterogeneous. The majority of interventions lack theoretical guidance, suffer high participant drop-out rates, and fail to describe how the intervention was adapted[21]. Whether culturally adapting DHIs contributes to a narrowing of the digital divide remains unclear. To fill this evidence gap, we propose prioritize a better understanding of existing challenges in designing and implementing cultural adaptations of DHIs, as well as identifying ways to mitigate them. This is a vital first step towards aligning best practice and informing successful and systematic approaches to culturally adapt DHIs.

## Critical challenges and mitigations

Although no DHI-specific frameworks exist to guide cultural adaptation of healthcare interventions, we believe the challenges generally fall into three categories: planning and designing, implementation, and technology (summarized in Box 1).

**Challenge 1: Defining culture in the context of health and technology**. Conducting a cultural adaption requires a clear definition of the culture and careful selection of relevant cultural aspects that are essential to include in a DHI. With over 100 definitions, culture is undeniably a multi-dimensional construct that goes well beyond ethnicity and race[9]. Culture, as defined by Castro et al., is the shared "worldviews and lifeways of a group of people"[8]. In the medical context, it includes cultural norms around therapies, the roles of healthcare professionals, care-related family traditions, the acceptance of medical technology, and many other context-specific variables. Each definition of culture has different adaptation implications. For example, defining culture with an emphasis on religious and spiritual norms may mean that the target group has deeply rooted and negative attitudes toward specific uses of technology (e.g., using smartphones to exchange pictures with healthcare providers of the opposite sex)[23]. On the other hand, defining culture focusing on community and family values might mean that it is essential to address social hierarchies and involve community leaders or family members to reach and engage people[23]. Understanding these concepts and then choosing which parts are essential for an adaptation are challenges that require cultural awareness, time, and close involvement of the targeted cultural group[8]. Ultimately, the adaptation team should include members (e.g., community advisory board) that know, fully understand, and live that culture, also understanding its views on health and technology.

**Challenge 2: Understanding and integrating sub-cultures.** Another challenge is the understanding and integration of

---

**Box 1 ❙ Critical challenges of cultural adaptations**

(1) Defining culture in the context of health and technology
(2) Understanding and integrating sub-cultures
(3) Justifying an adaptation
(4) Balancing planning, iteration, and stakeholder interests
(5) Ensuring intervention fidelity
(6) Understanding the interaction between health, technology, and culture

---

**Box 2 ❙ Mitigation recommendations**

(1) Weigh evidence, urgency, feasibility, and availability of resources
(2) Create a culturally aware and sensitive adaptation team
(3) Engage members of the target culture (e.g., community advisory board)
(4) Consult previous evidence, implementers of original DHI, and protocols
(5) Adapt stepwise, with multiple rounds of feedback
(6) Engage all relevant stakeholders, including technology experts

---

sub-cultural nuances. Seemingly homogeneous cultural groups consist of multiple subgroups with various degrees of cultural differences (sub-cultures), and thus, different understandings of health and technology[24]. Wainwright et al. used survey data to assess smoking and tobacco use among young Hispanic populations in the USA[25]. Their findings revealed that, compared to other Latino subgroups (e.g. Mexican and Dominican American), Cuban American respondents maintained more positive beliefs about smoking[25]. These were linked to cultural norms around social interactions, with Cuban Americans voicing stronger concerns about the negative social consequences of being a non-smoker[25]. Considering that tobacco, mainly in the form of cigars, is one of Cuba's main exports, strongly linked to its cultural heritage and national identity, these findings make sense[26]. If such differences exist, successful adaptations must disentangle and integrate these sub-cultural nuances. That includes knowing how to segment and engage these groups without drifting into impractical, overly-long, complicated, and costly adaptations. The adaptation team needs to be aware and sensitized to such nuances and how they might impact how a DHI is perceived. Engaging citizen scientists across these subgroups at the earliest adaptation stages may be one way to achieve that.

**Challenge 3: Justifying an adaptation**. Cultural adaptations come with risks and require money and time. An alternative to adaptations is the design of entirely new, culturally sensitive DHIs, which, however, comes with considerable cost demands[27]. Instead, adaptations aim for efficiency and adjust existing interventions. Yet, they come with risks because if not conducted properly, they might impose irrelevant concepts, leading to ineffective adaptations and wasted resources[28,29]. It is therefore key to justify an adaptation. Castro et al. suggest that before adapting, one of the following conditions should be met: (a) an intervention fails to engage specific cultural subgroups, (b) an intervention shows decreased efficacy for specific cultural groups, or (c) some cultural groups of interest have unique characteristics (such as risk factors, resilience factors, symptoms) linked to the intervention's outcomes[8]. While these are logical conditions, proceeding with an adaptation only after having enough evidence that an intervention is less efficacious for specific cultural groups is not always feasible. Most efficacy trials fail to include adequate cultural groups for this to be compared[30]. That leads to follow-up questions, such as: can one justify an adaptation without robust evidence that it is less efficacious for certain groups? And is an adaptation justified based only on efficacy or should there also be enough evidence of other differences in effectiveness (e.g., acceptability, feasibility) across cultural groups?[30] Ideally, these questions should be answered before an adaptation is conducted, considering several contextual factors such as the need for and urgency of a DHI, the feasibility of an adaptation and the availability of resources.

**Challenge 4: Balancing planning, iteration, and stakeholder interests**. Implementing a cultural adaptation requires a careful balance between planning, iterating, and collaborating. It also requires the involvement of multiple stakeholders, such as health experts, technology developers, community leaders, and members of the target group[8,31]. Bringing those stakeholders together and balancing their views and interests is a delicate yet essential task[31]. Those who conduct adaptations need to be culturally competent and attuned to how a DHI should be introduced into a community, as well as what impact it might have on its members[8,32]. They must also maintain awareness of their personal beliefs, values, and biases, as well as being mindful and open

to learning about other cultures that differ from their own[8,32]. Successful adaptations often involve multiple stages of stakeholder involvement (e.g., through interviews, focus groups, workshops), following iterative rounds of feedback and adaptations[33]. An example of this was the adaptation of the WHO's E-Mental Health Program for overseas Filipino workers where the first set of adaptations were based on feedback from Filipino psychologists. They suggested changes, such as removing images of doctors to avoid pathologizing mental health, known to be associated with stigma in Filipino culture. They then involved Filipino workers, followed by further iterative adaptation rounds[26,30]. Step-wise adaptation and feedback loops are important because correcting mistakes is inherently more expensive and time-consuming when technology is already set in place.

**Challenge 5: Ensuring intervention fidelity**. Maintaining the *fidelity* of a DHI, i.e., delivering the core intervention components as originally intended, is another implementation challenge[8,31,34]. Adapting too far or inappropriately (e.g., adding culturally unacceptable content) could reduce the efficacy of a DHI[8,31]. Cardemil considers this concern misplaced and argues that adaptations that target core components stop being adaptations and should be considered as entirely novel interventions[30]. The core components of a DHI often are the underlying behavior change techniques, the main messages that are delivered, and the technology used to deliver those[33]. Non-core and, therefore, adjustable elements often include its content language, tone, and layout (e.g., illustrations)[33]. It is vital to understand which digital elements of a DHI are considered core and which are not. Adjusting (e.g., changing the illustrations in a web platform) is costly and time-consuming and should be avoided if it alters a DHI's fidelity. The adaptation team should carefully consult previous publications, and those who implemented the original DHI, and, if available, follow existing protocols.

**Challenge 6: Understanding the interaction between health, technology, and culture**. Understanding how technology, health, and culture interact, as well as the target group's relationship with technology is an adaptation challenge unique to DHI. The links between health, technology, and cultural factors are bi-directional and inherently complicated. An essential part of understanding these links is a cultural group's relationship to technology, which includes beliefs, attitudes, and skills. A large-scale online survey by Lee et al. suggests that cultural elements related to uncertainty, individualism, contextuality, and time perception all impact that relationship[35]. At the same time, culture influences health perception and communication, determining how appropriate a technology is for a specific cultural group[33]. For example, a DHI for physical activity promotion based on a wearable without peer-sharing options might not be as effective in collectivist communities that value peer support and social validation. Similarly, a mobile health app with lenient data-sharing processes might not be appropriate to address sensitive topics (e.g., HIV prevention) within uncertainty-averse cultures that value transparency or privacy. Knowing how these are related is essential before choosing to adapt a DHI. Some cultural elements, such as family hierarchies, the importance of privacy, and gender roles render certain technologies, such as smartphones, a more suitable than other ways of delivering interventions[10]. To fully understand these linkages it is vital to consult an array of stakeholders, including community leaders, technology experts, and psychologists, in addition to the target audience.

## Outlook

In the DHI context, cultural adaptations remain an emerging field. Despite not being fully scientifically established yet, the potential to reach and engage traditionally underserved cultural subgroups and mitigate the digital divide is noteworthy. We believe that the field's future depends on how well we recognize and address critical challenges. Therefore, we call for realistic steps towards establishing DHI-specific frameworks on when and how to culturally adapt DHIs. These frameworks need to ensure that key concepts, such as digital literacy and trust are addressed in the context of culture. Future adaptations and their evaluations must acknowledge the digital divide at their core. Do adapted DHIs improve the reach of and engagement of underserved cultural groups? Can cultural adaptations help close the digital divide? To answer these questions, we need a better understanding of how cultural elements interact with technology, health, and digital inequities. Culture does not exist in a vacuum and should not be treated as such[36]. With non-acceptance and high drop-out rates being major DHI hurdles, that should be complemented with efforts to fill current evidence gaps on how adaptations impact DHI reach and engagement[37,38]. That includes exploring the interaction of cultural elements and different intervention delivery methods (e.g., smartphones, websites), and the impact of that interaction on reach and engagement of underserved cultural groups. Such efforts must be guided by methodological frameworks adjusted or developed specifically for DHIs. Frameworks should guide through the planning, design, and implementation stages while ensuring that all technology-specific considerations are met. Any choice to culturally adapt efficacious and effective DHIs should be made after carefully weighing available and missing evidence, ensuring that potential benefits outweigh likely risks.

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

## Acknowledgements

V.N. is funded by the Swiss National Science Foundation (Grant ID: 202914). Training support was provided to Sarah Jane Chavez (T32 AA007459, PI Monti).

## Author contributions

V.N. contributed to conceptualization, data curation, formal analysis, methodology, visualization, writing—original draft, and writing—review & editing. Paola Daniore contributed to validation, visualization, and writing—review & editing. Sarah Chavez contributed to validation, visualization, and writing—review & editing. Tyler Wray contributed to conceptualization, methodology, supervision, validation, visualization, and writing—review & editing.

## Competing interests

The authors declare no competing interests.
