## [Peer Review File · Communications Medicine]

Reviewers' comments:

Reviewer #1 (Remarks to the Author):

Overall I find this an interesting take on the role of culture in the potential to close the digital divide in healthcare/health interventions. I offer a few rather broad comments for consideration by the authors.

Summary (and then mentioned by the authors tangentially in lines 50-52): "Differences in the access and use of digital health interventions are driven by culture." I wholeheartedly agree, although it is not ONLY culture. Given the digital divide facing many lower income and/or rural households, I would add economic and physical access in addition to culture.

I appreciate the focus on the digital divide (up to line 66 or so). I wonder if it may be worth mentioning that for many who faced a lack of broadband internet but could access in public places – accessing healthcare is not necessarily feasible in that setting due to privacy concerns. I raise this only because it is often presented that public access sites (schools, public places providing free internet) can help to bridge the divide. That may be true, generally. But, healthcare and medicine seems a problematic match for public access points, in particular. Privacy is a key aspect for this topic in a way that it isn't for many (maybe most) other uses of technology.

Have the authors considered the type of device in the context of cultural acceptance? For example, is there potentially different acceptance for using a smartphone with an app to access care versus having to use a personal computer and/or to access via another device or online mechanism? I wonder only because there may be more acceptance for the use of mobile devices, like phones, compared to home computers which in addition to feeling more cumbersome, may simply be less accepted as tools for healthcare (whereas phones are already used to access care without the Internet capabilities).

Reviewer #2 (Remarks to the Author):

Interesting paper. This is an emerging and critical area of concern. The author has raised important issues around digital equity and implementation of interventions. Further analysis of the transferability of the ideas to USA demographics would be recommended. There is no statistical analysis. The ideas are general in nature and needs more specificity regarding actual results found or testable metrics.

I recommend publishing.

Reviewer #3 (Remarks to the Author):

At the end of the introduction, it is not clear what exactly the problem is and how this paper will address it. The objective is not clearly stated.

Unfortunately, the paper does not add much value and reliable insights to the existing literature.

Reviewer #4 (Remarks to the Author):

I would first like to congratulate the authors on submitting this perspective. Writing academic manuscripts takes time and getting it through to submission is in itself worthy of acknowledgment. Generally, the authors have a good writing style that is clear. The authors summarized current evidence on the cultural-digital divide, identified where the evidence was lacking, and then presented three phases that present challenges to culturally adapting DHIs.

Content comments

Each of the sections presented in this work is a brief overview and therefore not exhaustive. It is challenging to be exhaustive when there are stringent word limits for Perspectives. This is why I believe that Perspectives such be much more focused and narrower in scope so as to do justice to the topic. I have four major comments.

1. The authors should either focus on providing a more robust overview of the field and identifying evidence gaps OR providing more practical, actionable guidance of “when” and “how” to adapt DHIs or to develop frameworks to guide DHI development. The authors call for more work to be done by others in creating frameworks which is not a substantial contribution.
2. There needs to be some clarity on who is the audience of this piece. Is it researchers in cultural adaptations of DHI, or is it persons actually building and implementing the DHI? Can the authors clarify this in the summary and in the intro? Once clarified, it can help focus the work.
3. The bulk of the work in this Perspective is a limited review of the literature on each of the three challenges identified. More space should be allocated to this section which I see as a contribution to the literature. There should also be some clear, itemized actionable takeaways for each challenge. What is the intended audience supposed to do to address these challenges? What questions should be asked? Identifying challenges is not enough, they should provide some focused, actionable guidance. The authors propose some questions that can be asked but they come too late and only first appear in the Conclusion.
4. The section on ‘justified adaptations’ is a consideration for the design phase. This is when the developers need to make a decision on what to adapt and what not to. Therefore, it is unclear to me why this section is separate rather than integrated into the first challenge. For cohesiveness, I suggest moving this text into the ‘planning and design’ section.

Stylistic comments

A more explanatory title is needed. The “closing the divide?” The open-ended question doesn’t give enough, clear explanation of what this paper is about. What divide? Digital divide? Cultural-digital divide?

The use of open-ended questions as subtitles throughout the manuscript leaves much to be desired in explaining what is to come in the paragraph. I recommend that the authors consider more explanatory titles throughout.

Remark to authors and author responses:

Reviewer 1:

Overall I find this an interesting take on the role of culture in the potential to close the digital divide in healthcare/health interventions. I offer a few rather broad comments for consideration by the authors.

RESPONSE: Thanks a lot for your comments. Please find below our point-to-point replies to your concerns.

(1) Summary (and then mentioned by the authors tangentially in lines 50-52): “Differences in the access and use of digital health interventions are driven by culture.” I wholeheartedly agree, although it is not ONLY culture. Given the digital divide facing many lower income and/or rural households, I would add economic and physical access in addition to culture.

RESPONSE 1: Thanks for pointing that out. We changed the wording in the summary’s first sentence to reflect that (see p.2). No doubt, some of this digital divide is due to issues of access. However, there is also some evidence that DHIs may not reach or engage cultural minority groups as well as those in the majority culture. We added some sentences to highlight that in the introduction’s first paragraph (p.2) and in the paragraph “culture and the digital divide” (p.3).

(2) I appreciate the focus on the digital divide (up to line 66 or so). I wonder if it may be worth mentioning that for many who faced a lack of broadband internet but could access in public places – accessing healthcare is not necessarily feasible in that setting due to privacy concerns. I raise this only because it is often presented that public access sites (schools, public places providing free internet) can help to bridge the divide. That may be true, generally. But, healthcare and medicine seems a problematic match for public access points, in particular. Privacy is a key aspect for this topic in a way that it isn’t for many (maybe most) other uses of technology.

RESPONSE 2: We agree and added that aspect in the first paragraph of the introduction (p.2).

(3) Have the authors considered the type of device in the context of cultural acceptance? For example, is there potentially different acceptance for using a smartphone with an app to access care versus having to use a personal computer and/or to access via another device or online mechanism? I wonder only because there may be more acceptance for the use of mobile devices, like phones, compared to home computers which in addition to feeling more cumbersome, may simply be less accepted as tools for healthcare (whereas phones are already used to access care without the Internet capabilities).

RESPONSE 3: Thanks a lot for raising this important aspect. Due to space limitations, and the inherently narrow focus of a viewpoint, a detailed exploration of that aspect was not feasible. Yet we agree it is a very valid and important topic, that does not seem to have been properly answered by the scientific literature so far. We added a sentence under paragraph of challenge 6 (p.8).

Reviewer 2:

Interesting paper. This is an emerging and critical area of concern. The author has raised important issues around digital equity and implementation of interventions.

RESPONSE: Thanks a lot for your feedback. Please find below our point-to-point replies to your concerns.

(1) Further analysis of the transferability of the ideas to USA demographics would be recommended. There is no statistical analysis. The ideas are general in nature and need more specificity regarding actual results found or testable metrics.

I recommend publishing.

RESPONSE 1: Thank you very much for this remark. Considering that this is a viewpoint article, a statistical analysis or heavily citing quantitative results of studies was not considered appropriate (or feasible) as it would not allow for an easy-to-digest read. However, we fully acknowledge your concern and are already working on a full paper on the topic. This viewpoint merely aimed to provide an introductory discussion of an emerging area.

Reviewer 3: At the end of the introduction, it is not clear what exactly the problem is and how this paper will address it. The objective is not clearly stated. Unfortunately, the paper does not add much value and reliable insights to the existing literature.

RESPONSE: Thanks for taking the time to read our paper. We have revised and re-structured the entire viewpoint to make it clearer and easier to read.

Below are some of the changes we have made, that also address your concerns:

- (1) We revised the last paragraph of the introduction to summarize the viewpoint's main assumption, messages, and objective

(2) We have shifted some parts and start the viewpoint with (1) the linkages between culture and the digital divide, followed by (2) evidence gaps on the effectiveness of cultural adaptations of DHIs, and then (3) a list of key challenges and recommended mitigations of these challenges (mitigations/recommendations have been newly added)

We believe that this structure and new additions (e.g., recommendations) make for a valuable read and an important contribution to the literature, as that angle is currently lacking in the scientific literature.

Reviewer 4:

I would first like to congratulate the authors on submitting this perspective. Writing academic manuscripts takes time and getting it through to submission is in itself worthy of acknowledgment. Generally, the authors have a good writing style that is clear. The authors summarized current evidence on the cultural-digital divide, identified where the evidence was lacking, and then presented three phases that present challenges to culturally adapting DHIs.

RESPONSE: Many thanks for your kind remarks and constructive feedback. Below we address all your concerns.

Content comments

Each of the sections presented in this work is a brief overview and therefore not exhaustive. It is challenging to be exhaustive when there are stringent word limits for Perspectives. This is why I believe that Perspectives such be much more focused and narrower in scope so as to do justice to the topic.

I have four major comments.

(1) The authors should either focus on providing a more robust overview of the field and identifying evidence gaps OR providing more practical, actionable guidance of “when” and “how” to adapt DHIs or to develop frameworks to guide DHI development. The authors call for more work to be done by others in creating frameworks which is not a substantial contribution.

RESPONSE 1: We fully agree that a viewpoint cannot be too broad or exhaustive. We have therefore revised and restructured to focus the manuscript on the challenges. Attached to the challenges (usually towards the end of each paragraph) we have added some actionable

mitigation recommendations, which we summarize in panel 2. The viewpoint now starts with a very brief introduction into the linkages between culture and the digital divide (p.3), briefly mentions the current evidence gaps (p.4) and then transitions into the main part which are 6 key challenges and recommendations (p. 5-9). We have also added a new panel (mitigation recommendations) (p.9). To reflect these changes, we adjusted the manuscript title. We exchanged “closing the divide” with “challenges and mitigations”.

(2) There needs to be some clarity on who is the audience of this piece. Is it researchers in cultural adaptations of DHI, or is it persons actually building and implementing the DHI? Can the authors clarify this in the summary and in the intro? Once clarified, it can help focus the work.

RESPONSE 2: Thanks for pointing that gap out. Although it is written to address anyone interested in cultural adaptations and technology, the target audience is persons who aim to culturally adapt DHIs, which can be researchers, software developers (and others). We have clarified that in the last paragraph in the introduction. With that in mind, we believe that the narrower focus on (1) challenges and (2) recommendations is well aligned.

(3) The bulk of the work in this Perspective is a limited review of the literature on each of the three challenges identified. More space should be allocated to this section which I see as a contribution to the literature. There should also be some clear, itemized actionable takeaways for each challenge. What is the intended audience supposed to do to address these challenges? What questions should be asked? Identifying challenges is not enough, they should provide some focused, actionable guidance. The authors propose some questions that can be asked but they come too late and only first appear in the Conclusion.

RESPONSE 3: As mentioned in our response to one of your previous comments, we significantly shortened other parts of the paper, followed by some restructuring. Through that we have allocated most of the papers space to the six identified challenges and recommendations on how to mitigate those (p.5-9). This is not an exhaustive list, but what we believe (through personal experience and the literature) are the main points. The recommendations are also summarized in the newly added panel 2 (p.9). We have also revised the summary and conclusion paragraphs to reflect these changes.

(4) The section on 'justified adaptations' is a consideration for the design phase. This is when the developers need to make a decision on what to adapt and what not to. Therefore, it is unclear to me why this section is separate rather than integrated into the first challenge. For cohesiveness, I suggest moving this text into the 'planning and design' section.

RESPONSE 4: Thanks. We agree and have shifted that part into the challenges section (p.5-6).

Stylistic comments

(5) A more explanatory title is needed. The "closing the divide?" The open-ended question doesn't give enough, clear explanation of what this paper is about. What divide? Digital divide? Cultural-digital divide?

RESPONSE 5: We have revised the title to reflect the paper's narrower focus. We exchanged "closing the divide" with "challenges and mitigations". That title reflects the revisions made to the manuscript's stronger focus on challenges and mitigations.

(6) The use of open-ended questions as subtitles throughout the manuscript leaves much to be desired in explaining what is to come in the paragraph. I recommend that the authors consider more explanatory titles throughout.

RESPONSE: We have changed all sub-titles to more explanatory ones.

REVIEWERS' COMMENTS:

Reviewer #1 (Remarks to the Author):

Overall I find this a well-written manuscript which is clear and easily comprehended. I made several comments previously which I hoped to see reflected in the updated manuscript; all were. I appreciate the space limitations, but was glad to see that there was at least mention of the type of device used to access the Internet (I feel this is under-appreciated and important for this topic).

Reviewer #2 (Remarks to the Author):

What are the major claims of the paper?

The major claims are that cultural adaptation and better understanding of culture could potentially narrow the digital gap that exists for underserved populations.

Are they novel and will they be of interest to others in the community and the wider field?

Yes, this is a novel and needed approach to care delivery in potentially more culturally competent manner.

If the conclusions are not original, it would be helpful if you could provide relevant references.

This work is mostly original and while focused on the broader concept of DHI and cultural adaptation, a mention of digital redlining, trust and improving digital literacy should be addressed.

Is the work convincing, and if not, what further evidence would be required to strengthen the conclusions?

Yes, it is convincing. In my opinion, addressing root causes should be mentioned such as bias, racism, digital redlining and trust.

On a more subjective note, do you feel that the paper will influence thinking in the field? Yes, I believe this paper might motivate the digital health world. to better design telecare interventions. I would encourage the authors to take a look at the work of the American Telemedicine Association Disparity group.

Reviewer #3 (Remarks to the Author):

This revision has significantly improved the manuscript. Therefore, I have no further comments. I would recommend the manuscript for publication.

Point-to-point responses to reviewer remarks.

REVIEWERS' COMMENTS:

Reviewer #1 (Remarks to the Author):

Overall I find this a well-written manuscript which is clear and easily comprehended. I made several comments previously which I hoped to see reflected in the updated manuscript; all were. I appreciate the space limitations, but was glad to see that there was at least mention of the type of device used to access the Internet (I feel this is under-appreciated and important for this topic).

RESPONSE: Thank you very much for your comments and time.

Reviewer #2 (Remarks to the Author):

What are the major claims of the paper?

The major claims are that cultural adaptation and better understanding of culture could potentially narrow the digital gap that exists for underserved populations.

Are they novel and will they be of interest to others in the community and the wider field?

Yes, this is a novel and needed approach to care delivery in potentially more culturally competent manner.

If the conclusions are not original, it would be helpful if you could provide relevant references.

This work is mostly original and while focused on the broader concept of DHI and cultural adaptation, a mention of digital redlining, trust and improving digital literacy should be addressed.

RESPONSE: Thanks a lot. We added trust and digital literacy in the introduction and conclusions sections.

Is the work convincing, and if not, what further evidence would be required to strengthen the conclusions?

Yes, it is convincing. In my opinion, addressing root causes should be mentioned such as bias, racism, digital redlining and trust.

RESPONSE: Thanks a lot. We added bias, racism, and trust in the introduction.

On a more subjective note, do you feel that the paper will influence thinking in the field? Yes, I believe this paper might motivate the digital health world. to better design telecare

interventions. I would encourage the authors to take a look at the work of the American Telemedicine Association Disparity group.

RESPONSE: Thanks a lot for your comments and your time.

Reviewer #3 (Remarks to the Author):

This revision has significantly improved the manuscript. Therefore, I have no further comments. I would recommend the manuscript for publication.

RESPONSE: Thanks a lot for your comments and your time.